# A Theoretical Study of the Adsorption Process of B-aflatoxins Using *Pyracantha*
*koidzumii* (Hayata) Rehder Biomasses

**DOI:** 10.3390/toxins12050283

**Published:** 2020-04-28

**Authors:** Abraham Méndez-Albores, René Escobedo-González, Juan Manuel Aceves-Hernández, Perla García-Casillas, María Inés Nicolás-Vázquez, René Miranda-Ruvalcaba

**Affiliations:** 1UNAM–FESC, Campus 4, Multidisciplinary Research Unit L14 (Food, Mycotoxins and Mycotoxicosis), Cuautitlan Izcalli 54714, Mexico; albores@unam.mx; 2Department of industrial maintenance and nanotechnology, Technological University of Juarez City, Ciudad Juarez, Chihuahua 32695, Mexico; rene_escobedo@utcj.edu.mx; 3UNAM–FESC, Campus 1, Chemical Sciences Department, Cuautitlan Izcalli C. P. 54740, Mexico; juanmanuel.is.acevesh@gmail.com (J.M.A.-H.); mirruv@yahoo.com.mx (R.M.-R.); 4Institute of Engineering and Technology, Autonomous University of the City of Juarez, UACJ, City Juarez, Chihuahua 32584, Mexico; pegarcia@uacj.mx

**Keywords:** aflatoxin B_1_, biosorbents, theoretical studies, Density Functional Theory, B3LYP, quantum chemistry

## Abstract

Employing theoretical calculations with density functional theory (DFT) using the B3LYP/6-311++G(d,p) functional and basis set, the interaction of the aflatoxin B_1_ (AFB_1_) molecule and the functional groups present in the *Pyracantha koidzumii* biosorbent was investigated. Dissociation free energy and acidity equilibrium constant values were obtained theoretically both in solution (water) and gas phases. Additionally, the molecular electrostatic potential for the protonated molecules was calculated to verify the reactivity. Thus, methanol (hydroxyl group), methylammonium ion (amino group), acetate ion (carboxyl group), and acetone (carbonyl group), were used as representatives of the substrates present in the biomass; these references were considered using the corresponding protonated or unprotonated forms at a pH value of 5. The experimental infrared spectrophotometric data suggested the participation of these functional groups in the AFB_1_ biosorption process, indicating that the mechanism was dominated by electrostatic interactions between the charged functional groups and the positively charged AFB_1_ molecule. The theoretical determination indicated that the carboxylate ion provided the highest interaction energy with the AFB_1_ molecule. Consequently, an enriched biosorbent with compounds containing carboxyl groups could improve the yield of the AFB_1_ adsorption when using in vitro and in vivo trials.

## 1. Introduction

Aflatoxins (AFs) are biologically active secondary metabolites from certain fungal strains of *Aspergillus flavus* Link, *A. parasiticus* Speare and *A. nomius* Kurtzman et al. [1,2], being potentially fatal to humans and animals [3,4]. Aflatoxin B_1_ (AFB_1_), the most toxic structure of the four naturally occurring aflatoxins, is a direct-acting mutagen, being shown to disrupt genes involved in carcinogenesis and tumor suppression [5]. AFB_1_ also reacts in vivo with DNA to yield 2,3-dihydro-2-(N7-guanyl)-3-hydroxy AFB_1_ [6]. Reducing the consumption of aflatoxin-contaminated foods significantly minimizes the risk of an acute or chronic aflatoxicosis in humans; however, in developing countries such as Mexico, this scheme is not easy to achieve, due to the fact that the Mexican population has one of the highest world per capita consumption of maize, frequently contaminated with aflatoxins, both in the field as well as in storage [7].

Since the well-studied problem of aflatoxicosis in the early 1960s, researchers all over the globe have been searching for techniques to eliminate or to reduce the effects of these unavoidable food contaminants. However, due to the extensive variety of mycotoxins and their different chemical structures, protection against mycotoxicosis is a challenge task. Various methods have been investigated in connection with their effectiveness to control AF, with the objectives to either inactivate, to degrade, or to remove the mycotoxin, and they can be classified into physical, chemical, and biological [8,9,10,11,12,13]. Physical methods, such as binding adsorbents are by far the most practical and widely studied strategy for reducing the toxic effects of mycotoxin exposure. Recently, our research group proposed the use of a novel, natural, abundant, inexpensive, and effective AF binder based on *Pyracantha koidzumii* biosorbents [4]. Using an in vitro biosorption methodology, leaves and the mixture of leaves/berries presented the highest aflatoxin uptakes; 86% and 82%, respectively. Fourier transform infrared spectrophotometric studies suggested the participation of hydroxyl, carboxyl, and carbonyl groups in the biosorption process. Therefore, an adsorption model was proposed mainly by the electrostatic interactions between the negatively charged functional groups and the positively charged AF molecules.

Recently, computational theoretical chemistry methods have been used in property predictions, drug designs, and the study of molecular interactions in a wide variety of compounds [14,15,16,17]. In this context, a feasible strategy is the use of quantum chemical calculations to accurately explain the aflatoxin-biosorbent interactions and the main functional groups responsible in the biosorption process. Consequently, the objective of this work is to inform about the interactions of the AFB_1_ molecule with the different functional groups present in the *P. koidzumii* biosorbents in order to propose an enriched biosorbent to improve the yield of aflatoxin adsorption when using both in vitro an in vivo trials.

## 2. Results and Discussion

### 2.1. Study of the Protonation of Aflatoxin B_1_ (AFB_1_)

Recently, as a physical decontamination strategy, our research group reported the biosorption of the AFB_1_ molecule in an acidic medium by means of *Pyracantha koidzumii* biomasses (leaves, berries, and the mixture of leaves and berries in a 7:3 ratio) [4]. The biosorption model was proposed to be dominated by electrostatic interactions between the positively charged AFB_1_ molecules and the hydroxyl, ammonium, carboxylate and ketone-carbonyl groups of the biosorbent molecules.

In this research, a theoretical approach of the AFB_1_ biosorption is presented. Interaction between the AFB_1_ molecule and the functional groups of the biomass and identification of the main molecular groups involved in the biosorption process is studied. Since the experimental adsorption process was carried out in acidic conditions, the first step of this research was to analyze the AFB_1_ basic centers and their possible protonation structures (Figure 1).

In this regard, others authors have reported the protonation of the AFB_1_ molecule on the oxygen atom of the carbonylic groups (O_12_, O_14_), furan ring (O_7_), methoxyl group (O_13_) and the lactone ring (O_10_) [9,12,18], which is consistent with our findings. Additionally, a chemical structure with a hydrogen atom among the two carbonyl groups was considered (Figure 1A3,7). Moreover, two conformers were considered for the O_12_ and O_13_ protonated forms, one with intramolecular hydrogen bonding (O_12_∙∙∙H^+^∙∙∙O_13_) and two with the hydrogen attached only to one carbonyl group, O_12_∙∙∙H^+^ and H^+^∙∙∙O_13_. The chemical structures were optimized and evaluated by the acidity equilibrium constant both in gas and solution phases, in order to obtain their thermodynamic parameters. Isomeric structures were optimized at level of the Density Functional Theory (DFT), using calculations defined by the Becke’s three-parameter hybrid functional (B3LYP), employing the 6-311++G(d,p) basis set to obtain their molecular geometry and energy parameters after considering the solvent effect. The molecular geometry and some geometrical parameters for the structures are shown in Figure 2. Both the furan and lactone rings are not stabilized by the protonation process, yielding the opening of the corresponding ring, as previously reported, using basic and acidic conditions [10,11,13].

The structures A3–A7 exhibited the bond between the hydrogen atom and the oxygen atoms of the carbonyl groups with a length in the range of 0.971 to 0.989 Å, with A4 being the isomer with the shorter distance and the stronger bond, with a value of 0.971 Å, followed by A6 (0.973 Å). Additionally, two structures exhibited the hydrogen atom between the carbonyl groups (A3 and A7); these isomers established an intramolecular hydrogen bond. However, the structure with the shorter and stronger hydrogen bond was A7. The energy values of the different protonated chemical structures are summarized in Table 1. The results show that the structure with the carbonyl group, C=O_13_, and the O_12_- H^+^ intramolecular bond (Figure 2A3) is the most stable isomer, both in gas and solution phases. The isomer A4, with the hydrogen atom bond towards O_12_, but with the opposite orientation, has an energy difference of 3.31 and 7.96 kcal/mol in the water solution and gas phase, respectively.

The less stable structures were those with the hydrogen atom attached to the methoxyl group (A5), followed by the AFB_1_ structures with their protonated lactone (A2) or furan rings (A1). Thus, the instability of these structures could be attributed to the hydrolysis reaction carried out by the acidic medium. Thermodynamic parameters, such as Gibbs free energy of formation, were also determined at the same theory level and used to obtain the free energy required to protonate and the equilibrium constants. Results of the free energy and equilibrium constant values in the gas phase are summarized in Table 2.

Gibbs free energies indicate that, in the gas phase, the deprotonation process is spontaneous, showing a trend in the dissociation reaction to produce a neutral AFB_1_ molecule. Thus, pKa and acidity equilibrium constants showed higher values. In all cases, the deprotonated AFB_1_ molecule was the predominant species in equilibrium and the protonated species appeared to a lesser extent. Besides, the pKa values revealed that, in acidic medium and at the pH of the process, all AFB_1_ molecules were deprotonated. These results can be explained by considering the poor solvation effect in the gas phase and consequently the lower stability of the charged species. Additionally, the same calculations were carried out considering the solvent effect (water); results are shown in Table 3.

Interestingly, in the gas phase, the Gibbs free energy of formation for the protonated compounds showed lower negative values. However, in solution, structures with better stability were obtained when the AFB_1_ molecule was protonated, as a result of the solvation effect. Also, the dissociation free energies indicated that the loss of the hydrogen atom in the structures A1 (−30.9 kcal/mol), A2 (−37.8 kcal/mol), and A5 (−11.3 kcal/mol) was spontaneous (Table 2). However, the structures proposed for the O_12_ and O_13_ of the carbonyl groups (A3, A4, A6, and A7) revealed a nonspontaneous dissociation process with greater stability of the charged form in contrast with the neutral form. Moreover, the acidity constants and pKa values were obtained from the Gibbs free energy, showing positive or negative pKa values for A1 (0.17), A2 (−1.70), and A5 (−5.76), confirming that at the studied conditions of pH = 5, the structures were in a deprotonated form. On the other hand, the structures A3, A4, A6, and A7 showed greater pKa values at the above-mentioned pH value, indicating that these structures were in a protonated form, and consequently, involved in the biosorption process.

According to the previous results, the protonation study showed that only the structures A3, A4, A6, and A7 were in a protonated form under the experimental conditions tested, these structures have the oxygen atoms O_12_ and O_13_ attached to the carbonyl groups. Considering the Gibbs free energy, the most stable structures were A3 (−694,518.3 kcal/mol), A4 (−694,515.6 kcal/mol), A6 (−694,510.2 kcal/mol), and A7 (−694,513.4 kcal/mol). Structures A3 (O_13_∙∙∙HO_12_) and A7 (O_13_H∙∙∙O_12_) have the hydrogen ion between both carbonyl groups. Consequently, the interaction studies were made with the above-mentioned structures, when the hydrogen atom was attached to each carbonyl group with and without simultaneous interaction with O_12_ and O_13_. In general, analysis of the protonated AFB_1_ molecule allowed identifying co-existent structures and using them as models for the interaction study with the functional groups of the biosorbent. The experimental work proposed the AFB_1_ interaction with several molecular groups such as hydroxyl, ammonium ion, carboxylate ion, and ketone-carbonyl groups. Consequently, molecular models for these groups were—CH_3_OH for the hydroxyl group, CH_3_NH_3_^+^ for ammonium ion, and CH_3_COO^−^ for the carboxylate ion, and finally CH_3_COCH_3_ for the keto-carbonyl group. Furthermore, the protonated or deprotonated forms of the different molecular groups were also evaluated considering the experimental pH value; thus, the amino groups in the aminoacids and peptides are in the protonated form (ammonium ion); meanwhile, the carboxyl groups of the aminoacids, proteins, and other compounds are in deprotonated form (carboxylate ion).

### 2.2. Molecular Electrostatic Potential Surface

The molecular electrostatic potential (MEP) map is an advantageous property employed to predict appropriately the molecular reactivity of the compound of interest [19,20]. In other words, it is contemplated as an indicator of the reactivity regions of a target molecule, hence MEP has been employed in order to study the electron-donator and electron-acceptor interactions [21]. This property in the seven protonated forms of AFB_1_ was calculated at the B3LYP/6-311++G(d,p) level of theory, and it is indicated in a color range from +242 (deepest blue) to +146 (light blue). The corresponding maps are displayed in Figure 3. A positive potential value is concentrated in the vicinity of the hydrogen atom bonded to an oxygen atom. Therefore, interaction sites are proposed through hydrogen bridges. The AFB_1_H+ and hydroxyl, ammonium ion, carboxylate ion, and ketone-carbonyl groups of the biosorbent could act as both the proton donor and acceptors. The hydrogen atoms bonded the O_12_ (A3) and O_13_ (A7) oxygen atoms have a lower positive charge due to an intramolecular hydrogen bridge. This type of interaction causes a polarization through the AFB_1_ molecule, as observed in Figure 3. The MEP results indicate that the formation of the complexes depend on the protonation site of AFB_1_ molecule.

### 2.3. Studies of Protonated Aflatoxin B_1_ (AFB_1_H^+^) Interaction with Molecular Models

#### 2.3.1. Hydroxyl Groups: CH_3_OH

The principal interaction considered among the AFB_1_H^+^ molecule and the hydroxyl group was the hydrogen bond. Thus, various interactions with distinct conformations were considered and proposed between the AFB_1_ positively charged molecule and the hydroxyl groups. The hydrogen of the positive AFB_1_ molecule was oriented in two directions, first between both carbonyl groups (O_13_∙∙∙HO_12_ and O_13_H∙∙∙O_12_) and the opposite orientation (HO_12_ or HO_13_). These interactions are shown in Figure 4. The CH_3_OH interaction with the A3 and A7 structures shows two hydrogen bonds between the A3-OHCH_3_ yielding a strong hydrogen bond (1.968 Å and 128°) formed between the hydroxyl group of methanol and the O_13_ atom of the carbonyl group. The second hydrogen bond formed among O_12_ protonated with the oxygen atom of the hydroxyl group also leads to a very strong bond (1.390 Å and 170°) [22,23,24]. The interaction between A7 and CH_3_OH was also formed by two hydrogen bonds, one strong among the hydroxyl group as a donor and O_12_ as an acceptor (1.621 Å and 152.9°), and the other very strong bond between the protonated O_13_ and the CH_3_OH as an acceptor (1.281 Å and 180°). Moreover, A4 and A6 structures also interact with CH_3_OH (1.500–1.570 Å with angle of 170–176°) [25,26,27].

Concerning the energy parameters (Table 4), the most stable structures were observed in water dissolution instead of the gas phase; this energy difference was attributed to the solvation effect. The analysis in the gas phase revealed that the most stable interaction was A3-OHCH_3_ (−767,270.3 kcal/mol), while in solution it was A4-OHCH_3_ (−767,310.8 kcal/mol). In water, the energy difference among these structures was less than 1 kcal/mol (0.2 kcal/mol); however, in the gas phase, the difference was 5.2 kcal/mol. The energy difference between the highest and the lowest energy value in the gas phase was 10.5 kcal/mol. And the energy difference between the highest and the lowest energy value in the water phase was 5.0 kcal/mol.

The energy of interaction displayed a stronger hydrogen bond in the gas phase for all considered structures and a strong interaction in solution. It is important to note that the A6-OHCH3 and A7-OHCH3 structures have the greatest interaction energy in both phases, corresponding to a strong acidity constant. The correlation among the hydrogen bond energy and the acidity has been previously reported, pointing out that usually the strong acid is a better hydrogen donor and consequently increases the strength of the bond; however, this increment does not occur indefinitely [28,29]. In contrast, in the gas phase, a different behavior was observed, where the structure A7-CH3OH (−19.9 kcal/mol) showed the stronger interaction, followed by A6-CH3OH (−18.8 kcal/mol). Other researchers have indicated that, in gas phase, the strength of the hydrogen bond decreases corresponding to the basicity of two donor atoms; this phenomenon could be explained by the change in the trend in gas and solution phases [30,31,32,33].

#### 2.3.2. Ammonium Ion: CH_3_NH_3_^+^

The experimental work revealed a possible interaction with both the ammonium ion and the carboxylate ion; consequently, two molecular models were considered. CH_3_NH_3_^+^ was used for the ammonium ion and CH3COO^−^ for the carboxylate ion (Figure 5 and Figure 6). Both protonated (−NH_3_^+^) and un-protonated (−COO^−^) forms were defined by the pKa values previously reported [34,35]. In the interaction model, a carboxylate ion as an acceptor and an ammonium ion as a hydrogen donor are proposed.

The hydrogen bond length among the ammonium ion and the AFB_1_ molecule was in the range of 1.802 and 1.747 Å, which is in close agreement with previous reports (1.888−1.738 Å) [36]. In this context, shorter hydrogen bonds among ammonium ion and AFB_1_ carbonyl group (1.755 for A3-^+^H_3_NCH_3_ and 1.747 for A6-^+^H_3_NCH_3_) correspond to the structures with no intramolecular hydrogen bond, which can be explained by the electrostatic repulsion for the position of the hydrogen atom attached to the carbonyl group in the opposite side of the ammonium ion. In contrast, structures A3-^+^H_3_NCH_3_ and A7-^+^H_3_NCH_3_ have slightly larger hydrogen bond lengths (1.790 and 1.802, respectively) than the above-mentioned structures. These differences could be attributed to the additional hydrogen bond among the ketone-carbonyl groups (2.045 Å for A3-^+^H_3_NCH_3_ and 2.124 Å for A7-^+^H_3_NCH_3_). Thus, by increasing the electrostatic repulsion and consequently the bond length, the same phenomenon was observed for the hydrogen bond between the two carbonyl groups, showing an increase in the bond length in comparison with the structure without interaction. Regarding the energy of interaction in the gas phase (Table 5), the values indicate that all interactions were instable, with the least stable being A4-NH_3_^+^ (29.1099 kcal/mol). Therefore, the interactions in water solution were more stable for A3-^+^H_3_NCH_3_, A6-^+^H_3_NCH_3_, and A7-^+^H_3_NCH_3_. These structures have interaction energy values in the range of −1.2 to −9 kcal/mol, with A7-^+^H_3_NCH_3_ being the structure with the highest value (−8.6 kcal/mol).

#### 2.3.3. Carboxylate Ion: CH_3_COO^−^

Regarding the carboxylate ion, the association with the AFB_1_ protonated isomers was evaluated to establish the possible interaction sites, considering the unprotonated form of the acid. At the experimental pH, the AFB_1_ molecule is protonated and the carboxylic acid is unprotonated when they are in solution without interacting and when interacting, and the proton of the AFB_1_ molecule is transferred to the carboxylate ion.

In all interactions, the hydrogen atom of the protonated AFB_1_ was transferred to the carboxylate ion similarly to other carbonylic moieties (ketones, aldehydes, esters, and amides). Thus, the consequence of the primary effect of the neighboring OH is to increase the proton affinity of the C=O group [31,37,38,39,40]. The interaction model with the carboxylate ion is shown in Figure 6. The structures A4-^−^OOCCH_3_ and A6-^−^OOCCH_3_ have a strong hydrogen bond (1.745 and 1.821 Å, with angles of 173.6° and 167.2°, respectively); meanwhile, the structures A3-^−^OOCCH_3_ and A7-^−^OOCCH_3_ have two hydrogen bonds, one stronger (1.821 and 1.809 Å, respectively), and the second weaker (2.765 and 2.719 Å, respectively), with bond angles of 110.9° and 114.9°, respectively.

In the gas phase (Table 6), a stronger interaction in comparison to that due to water solvation is shown as a consequence to the ionic nature species bonding and the lower stabilization effect. A neutral structure generates the most stable structure in the medium as a result of the charge attraction. However, the solvation effect that stabilizes the polar molecules and ionic structures also improves the dispersion effect and reduces the interaction energy (Table 6). In general, the most stable structure was A4-^−^OOCCH_3_ (−838,181.1 kcal/mol in the gas phase, and −838,198.0 kcal/mol in water solution). In water, the best interaction energy corresponded to A6-^−^OOCCH_3_ (−40.2 kcal/mol), followed by A7-^−^OOCCH3 (−36.0 kcal/mol). Finally, the structure with the lowest interaction energy was A4-^−^OOCCH_3_ (−31.69 kcal/mol).

#### 2.3.4. Carbonyl Group: (CH_3_)_2_C=O

The carbonyl group involved in the AFB_1_ adsorption is also seen in peptides, aminoacids, carbohydrates, and some secondary metabolites present in the biosorbent [34]. The interaction among this group and the protonated AFB_1_ molecule was studied using the (CH_3_)_2_C=O molecule as a model, which can act as a hydrogen bond acceptor [41]. These interactions are depicted in Figure 7. Structures showed the hydrogen bond between the carbonyl group of the (CH_3_)_2_C=O and the protonated AFB_1_ molecule. The interaction model with structures A3-O=C(CH_3_)_2_ and A7-O=C(CH_3_)_2_ exhibited two hydrogen bonds, the first weaker, among the two carbonyl groups of the AFB_1_ molecule (2.579 Å and 111.5° for A3-O=C(CH_3_)_2_, and 2.600 Å and 94.6° for A7-O=C(CH_3_)_2_), and a stronger hydrogen bond with the O=C(CH_3_)_2_ (1.461 Å and 160.1° for A3-O=C(CH_3_)_2_ and 1.411 Å and 168.9° for A7-O=C(CH_3_)_2_). Structures A4-O=C(CH_3_)_2_ and A6-O=C(CH_3_)_2_ revealed only one stronger hydrogen bond with the carbonyl model, with geometrical parameters of 1.524 Å and 179.3° for A4-O=C(CH_3_)_2_, and 1.453 Å and 174.4° for A6-O=C(CH_3_)_2_, respectively.

With regard to the interaction energy (Table 7), it was stronger in the gas phase than that in solution phase, which is consistent with other results obtained in this work. Furthermore, the interaction energy among the structures A4-O=C(CH_3_)_2_ and A6-O=C(CH_3_)_2_ (−7.3 and −11.4 kcal/mol, respectively) was greater than that of A3-O=C(CH_3_)_2_ and A7-O=C(CH_3_)_2_ (−5.2 and −5.4 kcal/mol, respectively), which was attributed to the second hydrogen bond in the last two structures, causing a weak interaction with the carbonyl model.

Based on these results, the best interaction was obtained with the carboxylic groups; consequently, carboxylate ion-enriched biosorbents would be the best option for AFB_1_ adsorption with the experimental conditions tested.

### 2.4. Infrared Spectrophotometric Studies of the Interactions

Complementary and interesting results were obtained for the theoretical infrared spectrum calculations at the same theory level, searching for experimental band shifts. The infrared spectra of the molecule models are shown in Figure 8. The predicted infrared spectrum showed bands of hydroxyl groups above 3800 cm^−1^ and amine groups at 3500 cm^−1^, both bands in the experimental spectrum were broader [41], mainly due to the hydrogen bond formation; however, the theoretical calculation was made on single molecules; therefore, the bands in the predicted spectrum were very sharp. In all cases, alkyl chain bands were around 3000 cm^−1^; also, the carbonyl group bands of ketone and carboxylate ions were found at 1785 and 1648 cm^−1^, respectively. Finally, the bands of N-H deformation appeared between 1600 and 1500 cm^−1^. In addition, the infrared spectrums of the protonated AFB_1_ molecules (A3, A4, A6, and A7) were predicted and are shown in Figure 9. All data are in close agreement to previously reported results [41].

The infrared spectra showed bands at 3700–3300 cm^−1^ attributable to the stretching of the hydrogen- carbonyl group bond, located at low wavenumbers when an intramolecular hydrogen bond was formed as a consequence of the minor energy required by the bond vibration [42,43]. Other intense bands were assigned to the stretch of carbonyl groups (1739 to 1800 cm^−1^), and to the symmetrical and nonsymmetrical stretching of the carbon-carbon double bonds (1500–1600 cm^−1^). Results are in close agreement with the experimental data found in the AFB_1_ infrared spectrum [4,12,44,45,46]. Interestingly, structures A3 and A6 have similar patterns to those of the experimental spectrum. Moreover, the biosorbent-aflatoxin interactions were calculated and compared with the experimental results previously reported [4] in order to confirm the proposed changes. The obtained results are displayed in Figure 10, Figure 11, Figure 12 and Figure 13.

Theoretical infrared spectrum bands showed the hydroxyl group vibration band toward less wavenumbers [4]. This variation was associated with the hydrogen bond formation and consequently the lower stretching energy for the O-H bond. The CH_3_OH interaction with A7 was similar to the experimental results. Additionally, the hydrogen bond formed with the ammonium ion also exhibits a shift to lower wavenumbers and lower energy values. Regarding the carboxyl group, the bands corresponding to the hydroxyl group showed a shift to lower wavenumbers appearing at around 3400 cm^−1^. Also, the carbonyl group changes the wavenumber from 1648 to around 1700 cm^−1^, as a consequence of the hydrogen bond interaction with the lactone ring of the AFB_1_ molecule. Additionally, the carbonyl group revealed a reduction in its infrared activity with a greater reduction in the A6 structure. Finally, theoretical spectrophotometric results of the carbonyl group showed the corresponding intense band of the hydrogen bond among the AFB_1_ and the employed models (wavenumbers between 2100 to 2700 cm^−1^). The nonprotonated carbonyl group appeared at around 1800 cm^−1^, while the protonated group was detected around 1600 to 1700 cm^−1^. Thus, the offered theoretical infrared information contributes to confirm the feasibility of the interaction model proposed in the present work.

## 3. Conclusions

A theoretical study of the AFB_1_-biosorbent interactions was carried out. Acidity constants of some protonated structures of the AFB_1_ molecule were calculated. The protonated A3 molecule was the most stable structure. The protonated AFB_1_ structures that are in the 13 kcal/mol range were A3, A4, A6, and A7. Structures A1, A5, and A7 were found to be less stable, due to protonated oxygen atoms in both furan and lactone rings. The interaction energy increases in the order of methylammonium ions (amino group) < acetone (carbonyl group) < methanol (hydroxyl group) < acetate ions (carboxyl group). Consequently, carboxylate ion-enriched biosorbents would be the best option for AFB_1_ adsorption in both in vitro and in situ trials. Research in this direction is in progress.

## 4. Materials and Methods

### 4.1. Optimization of the Structures

This study considered the aflatoxin molecule maximum stability stereoisomer previously reported by our research group [10,47]. In the first stage, the interactions were built with standard bond lengths and bond angles, using the PC Spartan06 program [48]. Therefore, the first task was to establish the conformation of maximum stability. Thus, the geometry for each isomer was fully optimized using DFT calculations, which were obtained using the Gaussian 09 program [49]. These calculations were carried out defined by the Becke’s three parameter hybrid functional (B3LYP) [50,51], which include a mixture of Hartree–Fock exchange with DFT exchange correlation. The used basis set includes the split-valance and diffuse functions, 6-311++G(d,p) [52,53,54,55,56,57]. The substrate model optimizations were carried out considering the protonated or unprotonated form at the experimentally studied pH.

### 4.2. Thermochemical Parameters and Acidity of the Structure and Models

Thermochemical values were estimated from frequency calculations, which included a thermochemical analysis of the system considering 298.15 K, 1 atm of pressure and the principal isotope for each element [39,58,59], all at the same theory level. Zero-point correction to the electronic energy (ZPE) of the molecule was used to calculate the values of enthalpy and Gibbs free energy. The solvent effect was also calculated by using the self-consistent reaction field (SCRF) method and considering the Tomasi’s polarizable continuum model (PCM) at the same theory level, using water as a medium. The procedure to calculate solvation model involved first optimizing the structure in the gas phase and then again optimizing the optimized structure for the solvation model. For all energies, ZPE corrections were taken into consideration [60,61]. The acidity constants were calculated using the relationship between free energy and the equilibrium constants using the following equation [62]:Δ*G°* = −*RTLnkeq*(1)
where Δ*G°* was the Gibbs free energy of the deprotonation process, calculated by the difference among the formation free energy of products and the reactants. The *keq* values of the deprotonation process were used as acidity constants.

### 4.3. Molecular Electrostatic Surface

The molecular electrostatic potential (MEP) map is often used as a powerful predictive and interpretative tool in fields such as chemistry and intermolecular interactions. MEP is widely used to make a reactivity map displaying the molecular regions most probable to get the electrophilic attack of reagents in the molecules [63]. It maps potentials created in the space around a molecule by its nuclei and electrons. This also provides information for understanding the shape, size, charge density, delocalization and site of chemical reactivity of the molecules. MEP can be measured experimentally by diffraction as well as computed by quantum mechanical methods. MEP maps can be obtained by mapping electrostatic potential onto the total electron density with a color code [64]. MEP contour maps provide a simple way to predict how different geometries could interact [65]. This property was determined using the DFT (B3LYP/6-311++G(d,p) method.

### 4.4. Vibrational Analysis

Density functional theory (DFT) calculations are reported to provide vibrational frequencies of organic compounds [66,67,68,69,70,71,72]. Consequently, in this work, by using the DFT (B3LYP/6-311++G(d,p) method, the respective vibrational frequencies were calculated. The outcome represents the vibrational force constants of the molecule and the vibrational frequencies of vibrational modes and the integrated intensities of the infrared bands. Therefore, these calculations are valuable to provide insight into the vibrational spectrum. According to Rui-Zhou et al. [72], the B3LYP function yields a good description of harmonic vibrational wavenumbers for small- and medium-sized molecules; the calculations refer to the isolated molecules while the experimental vibrational spectra, as previously reported, were recorded mostly in condensed phase.

### 4.5. Interaction Energy Calculations

The association energies through the hydrogen bond shown in the AFB_1_–biosorbent interaction were calculated by subtracting the energy of the isolated molecules from the total energy of the complex [14,15,16].

## Figures and Tables

**Figure 1 toxins-12-00283-f001:**
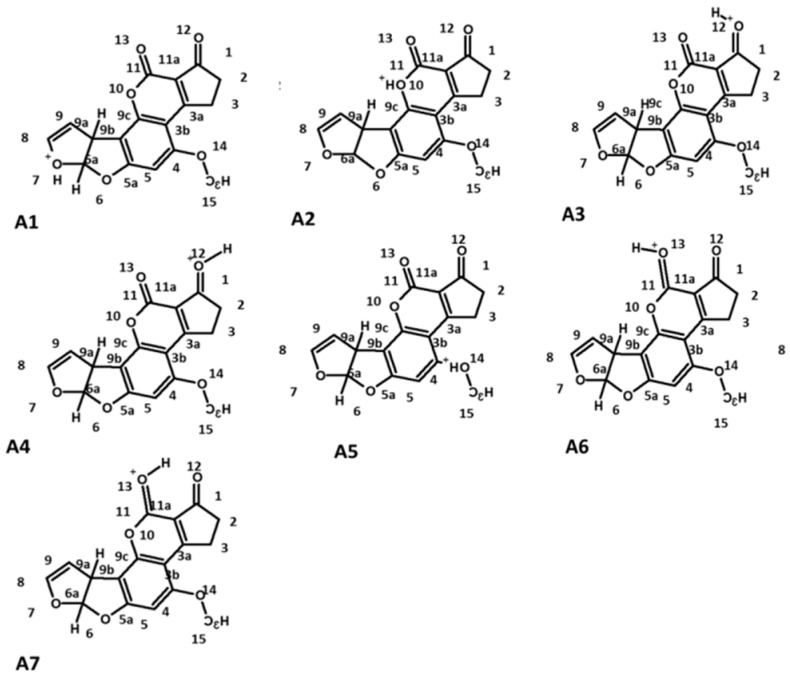
Proposed structures of the protonated aflatoxin B_1_ (AFB_1_) molecule. **A1** (AFB_1_O_7_-H+), **A2** (AFB_1_O_10_-H+), **A3** (AFB_1_O_13_O_12_-H+), **A4** (AFB_1_O_12_-H+), **A5** (AFB_1_O_14_-H+), **A6** (AFB_1_O_13_-H+), and **A7** (AFB_1_O_12_-O_13_-H+).

**Figure 2 toxins-12-00283-f002:**
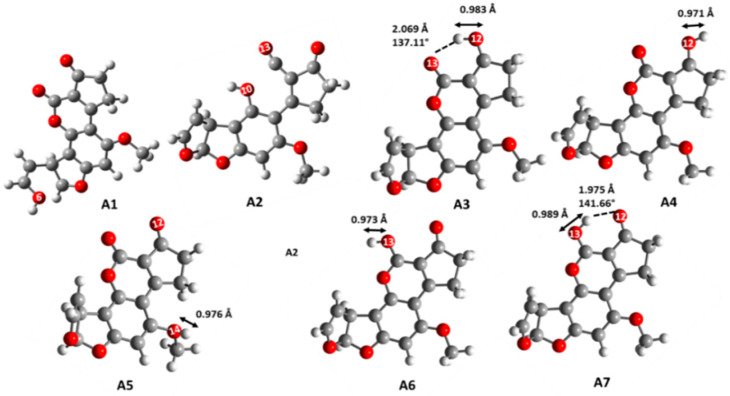
Optimized structures of the positively charged AFB_1_ molecules A1–A7. **A1** (AFB_1_O_7_-H+), **A2** (AFB_1_O_10_-H+), **A3** (AFB_1_O_13_O_12_-H+), **A4** (AFB_1_O_12_-H+), **A5** (AFB_1_O_14_-H+), **A6** (AFB_1_O_13_-H+), **A7** (AFB_1_O_12_-O_13_-H+).

**Figure 3 toxins-12-00283-f003:**
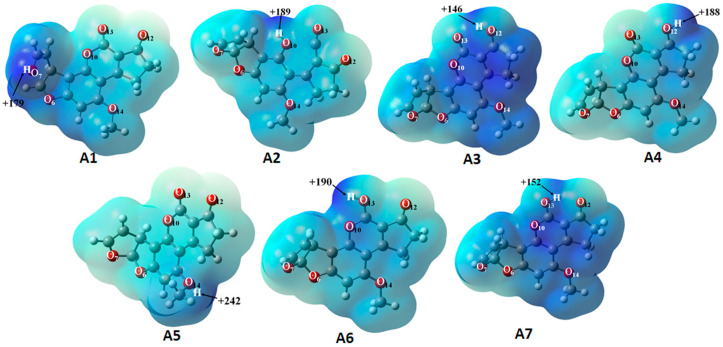
The total electron density isosurface mapped with molecular electrostatic potential of—**A1** (AFB_1_O_7_-H^+^), **A2** (AFB_1_O_10_-H^+^), **A3** (AFB_1_O_13_O_12_-H^+^), **A4** (AFB_1_O_12_-H^+^), **A5** (AFB_1_O_14_-H^+^), **A6** (AFB_1_O_13_-H^+^), and **A7** (AFB_1_O_12_-O_13_-H^+^). Density = 0.0004 and isovalue = 0.02. Level of calculation: B3LYP/6-311++G(d,p). The color scheme for the MEPS is as follows—blue, electron-deficient, partially positive charge; light blue, slightly electron-deficient region.

**Figure 4 toxins-12-00283-f004:**
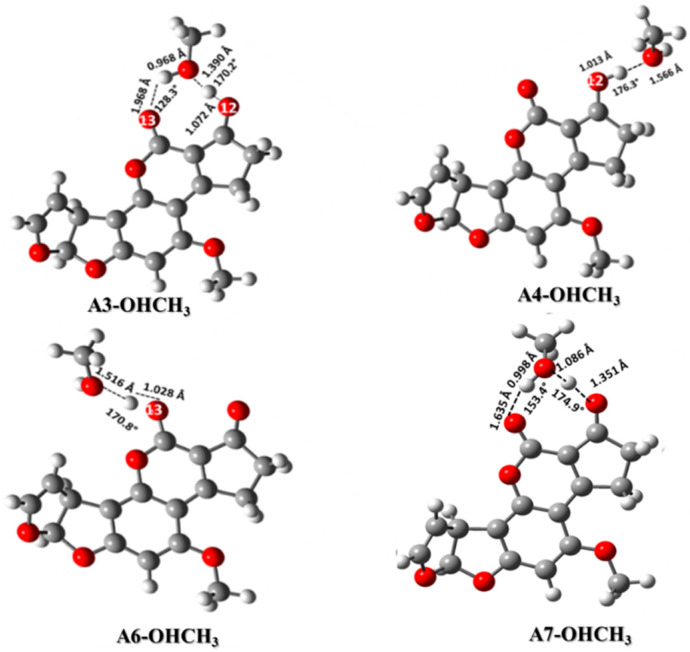
Interaction of the AFB_1_ protonated forms with the hydroxyl group using CH_3_OH as a model system. A3-OHCH_3_ (AFB_1_-O_13_O_12_-H^+^∙∙∙OHCH_3_), A4-OHCH_3_ (AFB_1_O_12_H^+^∙∙∙OHCH_3_), A6-OHCH_3_ (AFB_1_O_13_H^+^∙∙∙OHCH_3_), and A7-OHCH_3_ (AFB_1_O_12_-O_13_-H^+^∙∙∙OHCH_3_).

**Figure 5 toxins-12-00283-f005:**
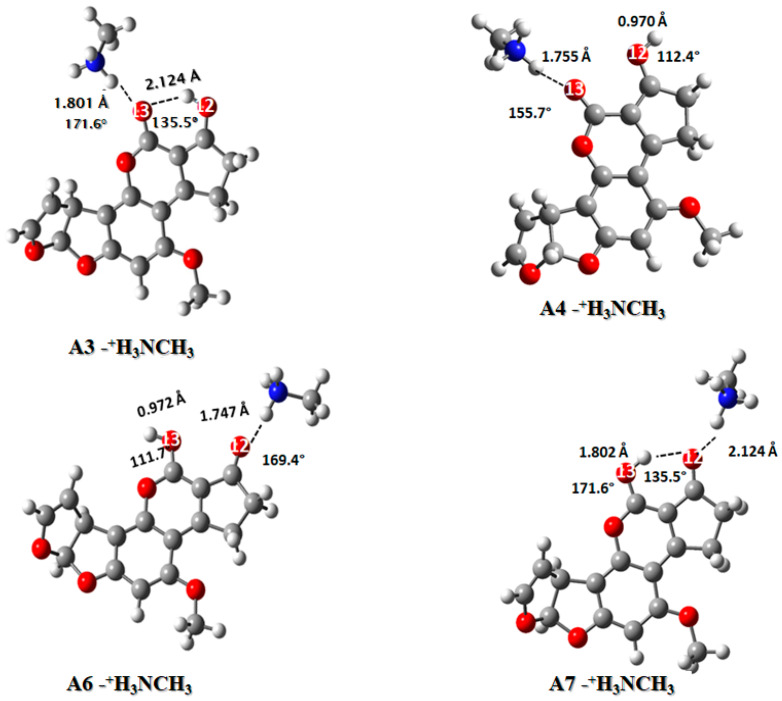
Interaction of the AFB_1_ protonated forms with the protonated amine group, CH_3_NH_3_^+^ as a model. A3-^+^H_3_NCH_3_ (AFB_1_O_12_O_13_∙∙∙^+^H_3_NCH_3_), A4-^+^H_3_NCH_3_ (AFB_1_O13∙∙∙^+^H_3_NCH_3_), A6-^+^H_3_NCH_3_ (AFB_1_O_12_∙∙∙^+^H_3_NCH_3_) and A7-^+^H_3_NCH_3_ (AFB_1_O_13_O_12_∙∙∙^+^H_3_NCH_3_).

**Figure 6 toxins-12-00283-f006:**
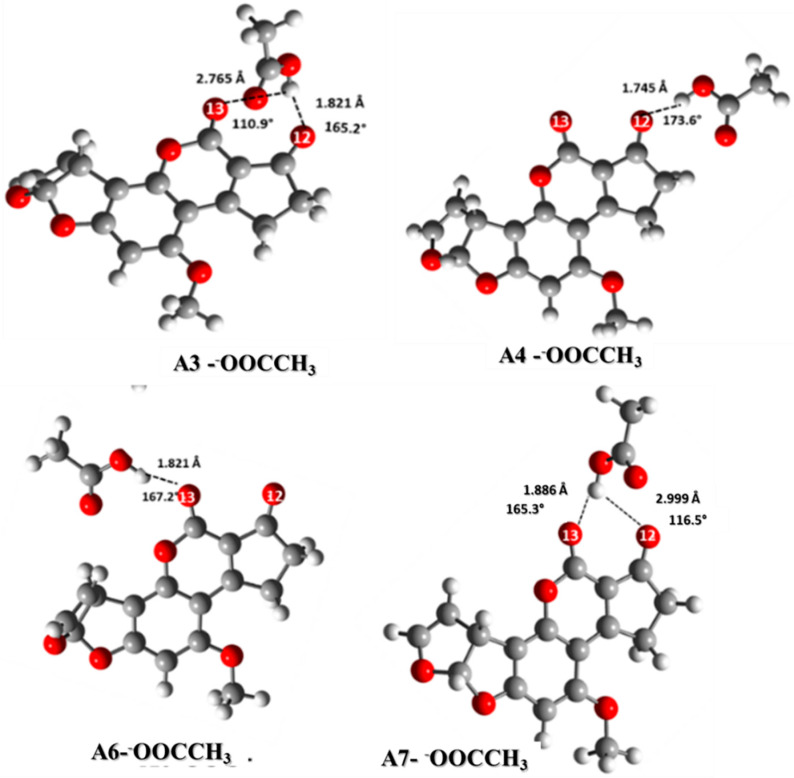
Interaction of the AFB_1_H^+^ with the carboxylate ion using CH3COO^−^ as a model system. A3-^−^OOCCH_3_ (AFB_1_O_13_O_12_-H^+^∙∙∙^−^OOCCH_3_), A4-^−^OOCCH_3_ (AFB_1_O_12_H^+^∙∙∙^−^OOCCH_3_), A6-^−^OOCCH_3_ (AFB_1_O_13_H^+^∙∙∙^−^OOCCH_3_) and A7-^−^OOCCH_3_ (AFB_1_O_12_O_13_-H^+^∙∙∙^−^OOCCH_3_).

**Figure 7 toxins-12-00283-f007:**
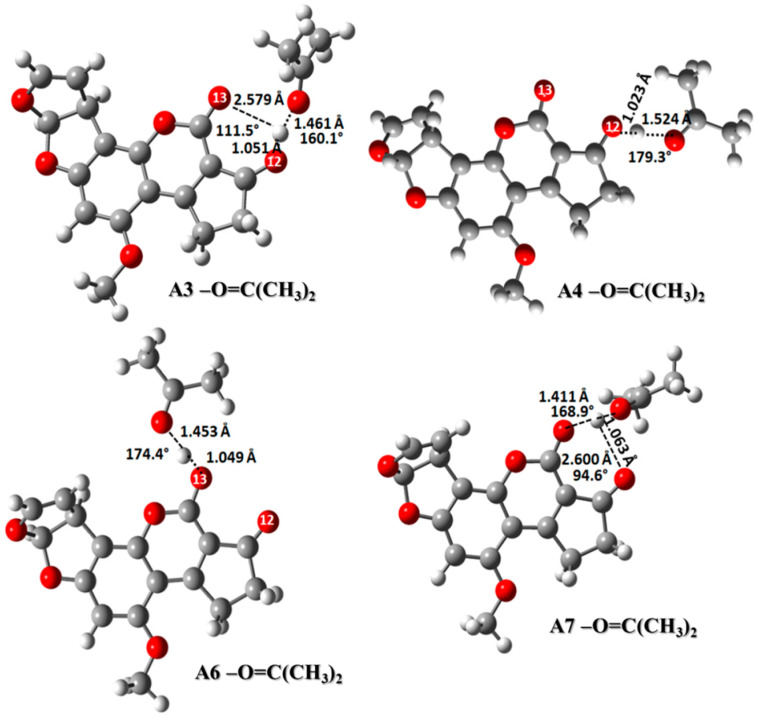
Interaction of the AFB_1_ protonated forms with the carbonyl group using O=C(CH_3_)_2_ as a model system. A3-O=C(CH_3_)_2_ (AFB_1_O_13_O_12_-H^+^∙∙∙O=C(CH_3_)_2_), A4-O=C(CH_3_)_2_ AFB_1_O_12_-H^+^∙∙∙O=C(CH_3_)_2_), A6-O=C(CH_3_)_2_ (AFB_1_O_13_-H^+^∙∙∙O=C(CH_3_)_2_) and A7-O=C(CH_3_)_2_ AFB_1_O_12_O_13_-H^+^∙∙∙O=C(CH_3_)_2_).

**Figure 8 toxins-12-00283-f008:**
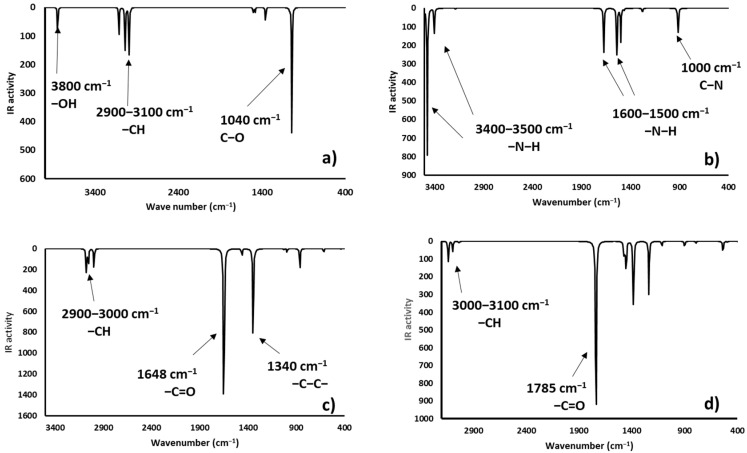
Predicted infrared spectra (without correction factor) for the model systems—(**a**) hydroxyl (methanol), (**b**) ammonium ion (methylamine), (**c**) carboxylate ion (acetic acid), and (**d**) carbonyl group (acetone).

**Figure 9 toxins-12-00283-f009:**
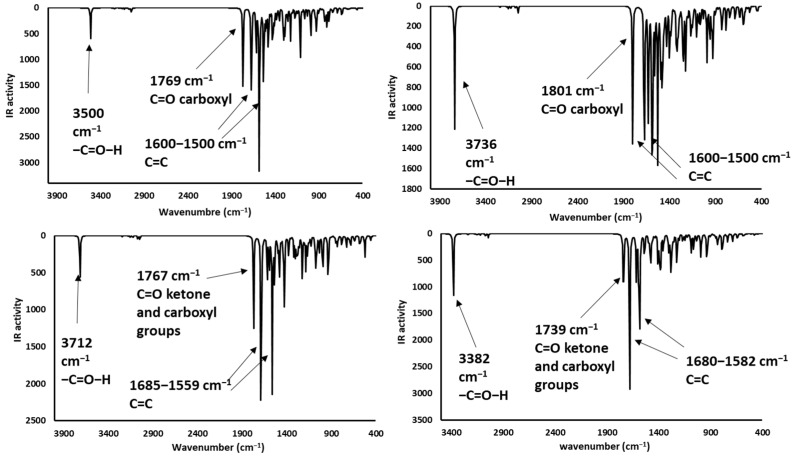
Theoretical infrared spectrums of the protonated AFB_1_ molecules—(**a**) **A3** (AFB_1_O_13_O_12_-H^+^), (**b**) **A4** (AFB_1_O_12_-H^+^), (**c**) **A6** (AFB1O_13_-H^+^), and (**d**) **A7** (AFB_1_O_12_-O_13_-H^+^).

**Figure 10 toxins-12-00283-f010:**
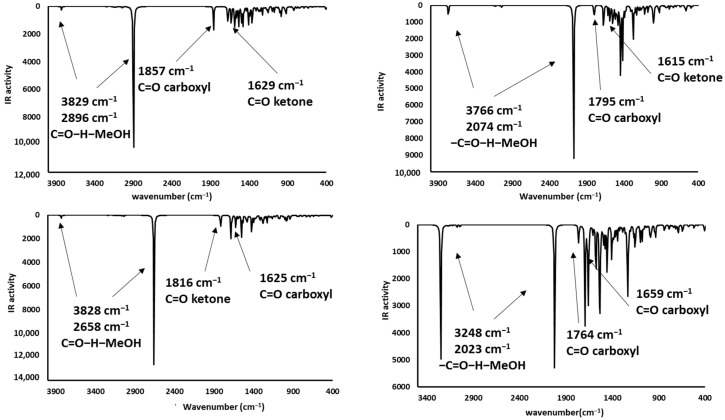
Theoretical infrared spectra of CH_3_OH interactions—(**a**) **A3** (AFB_1_-O_13_O_12_-H^+^∙∙∙OHCH_3_), (**b**) **A4** (AFB_1_O_12_H^+^∙∙∙OHCH_3_), (**c**) **A6** (AFB_1_O_13_H^+^∙∙∙OHCH_3_) and (**d**) **A7** (AFB_1_-O_12_O_13_-H^+^∙∙∙OHCH_3_).

**Figure 11 toxins-12-00283-f011:**
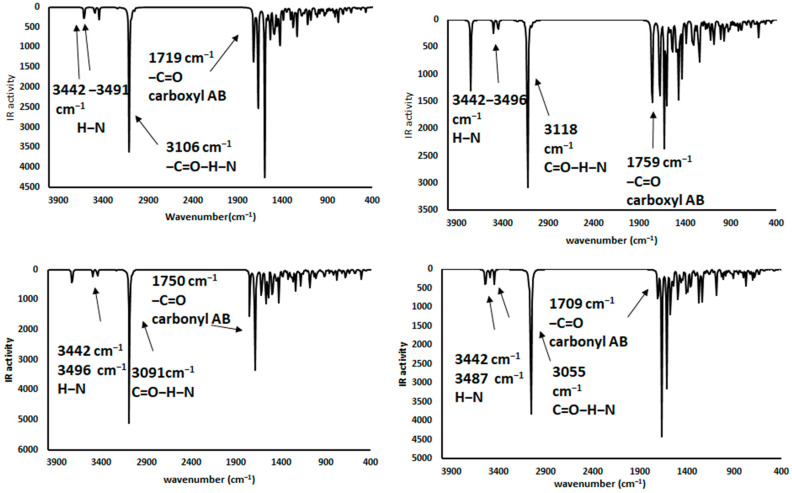
Theoretical infrared spectra of CH_3_NH_3_^+^ interactions—(**a**) **A3** (AFB_1_O_12_O_13_∙∙∙^+^H_3_NCH_3_), (**b**) **A4** (AFB_1_O_13_∙∙∙^+^H_3_NCH_3_), (**c**) **A6** (AFB_1_O_12_∙∙∙^+^H_3_NCH_3_) and (**d**) **A7** (AFB_1_O_13_O_12_∙∙∙^+^H_3_NCH_3_).

**Figure 12 toxins-12-00283-f012:**
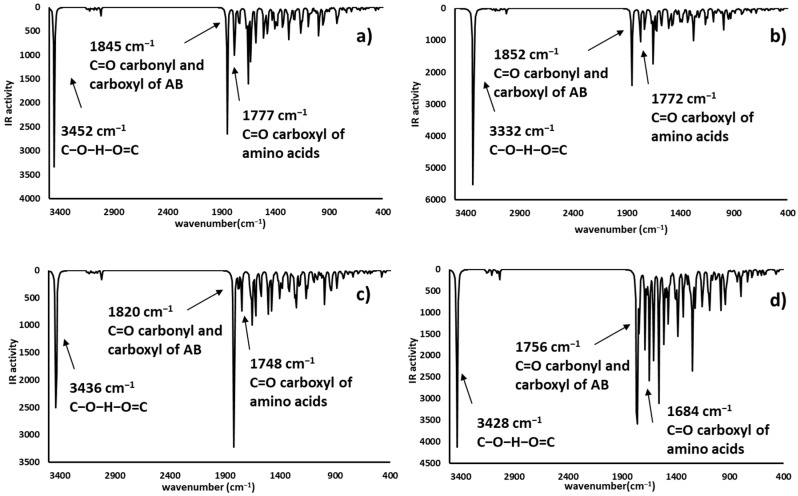
Theoretical infrared spectra of CH_3_COO^−^ interactions—(a) **A3** (AFB_1_O_13_O_12_-H^+^∙∙∙^−^OOCCH_3_), (**b**) **A4** (AFB_1_O_12_H^+^∙∙∙^−^OOCCH_3_), (**c**) **A6** (AFB_1_O_13_H^+^∙∙∙^−^OOCCH_3_) and (**d**) **A7** (AFB_1_O_12_O_13_-H^+^∙∙∙^−^OOCCH_3_).

**Figure 13 toxins-12-00283-f013:**
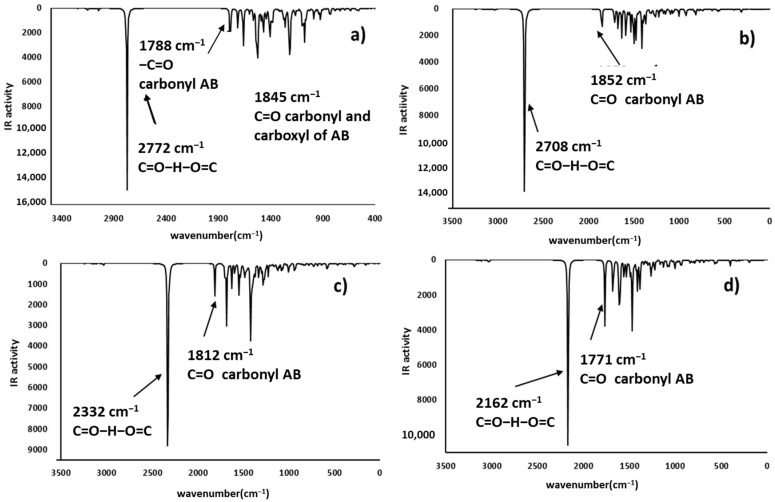
Theoretical infrared spectra of (CH_3_)_2_CO interactions—(**a**) **A3** (AFB_1_O_13_O_12_-H^+^∙∙∙O=C(CH_3_)_2_), (**b**) **A4** (AFB_1_O_12_-H^+^∙∙∙O=C(CH_3_)_2_), (**c**) **A6** (AFB_1_O_13_-H^+^∙∙∙O=C(CH_3_)_2_) and (**d**) **A7** (AFB_1_O_12_O_13_-H^+^∙∙∙O=C(CH_3_)_2_).

**Table 1 toxins-12-00283-t001:** Energy values of the protonated structures of the AFB_1_ optimized molecule. **A1** (AFB_1_O_7_-H^+^), **A2** (AFB_1_O_10_-H^+^), **A3** (AFB_1_O_13_O_12_-H^+^), **A4** (AFB_1_O_12_-H^+^), **A5** (AFB_1_O_14_-H^+^), **A6** (AFB_1_O_13_-H^+^), **A7** (AFB_1_O_12_-O_13_-H^+^).

Protonated Structures AFB_1_	Water Solution		Gas Phase	
	Energy Kcal/mol	ΔE Kcal/mol	Energy Kcal/mol	ΔE Kcal/mol
**A1**	−694,634.98	25.91	−694,577.79	38.45
**A2**	−694,633.57	27.32	−694,585.85	30.38
**A3**	−694,660.89	0	−694,616.24	0
**A4**	−694,657.58	3.31	−694,608.31	7.96
**A5**	−694,629.32	31.57	−694,560.57	55.66
**A6**	−694,651.91	8.98	−694,602.81	13.43
**A7**	−694,655.76	5.14	−694,611.20	5.03

ΔE = E_energy**A1…A7**_ − E_energy**A3**_.

**Table 2 toxins-12-00283-t002:** Dissociation free energy and acidity equilibrium constant values in the gas phase for—**A1** (AFB_1_O_7_-H^+^), **A2** (AFB_1_O_10_-H^+^), **A3** (AFB_1_O_13_O_12_-H^+^), **A4** (AFB_1_O_12_-H^+^), **A5** (AFB_1_O_14_-H^+^), **A6** (AFB_1_O_13_-H^+^), **A7** (AFB_1_O_12_-O_13_-H^+^).

AFB_1_H^+^ ⇆ AFB_1_ + H^+^				
Structure	ΔG Formation kcal/mol	ΔG Dissociation Reaction kcal/mol	Ka	pKa
**A1**	−694,440.146	−30.914	4.575 × 10^22^	−22.661
**A2**	−694,447.074	−37.843	5.483 × 10^27^	−27.739
**A3**	−694,473.686	−64.454	1.759 × 10^47^	−47.245
**A4**	−694,466.299	−57.067	6.767 × 10^41^	−41.830
**A5**	−694,420.522	−11.291	1.888 × 10^8^	−8.276
**A6**	−694,461.097	−51.865	1.041 × 10^38^	−38.018
**A7**	−694,468.856	−59.625	5.071 × 10^43^	−43.705

**Table 3 toxins-12-00283-t003:** Dissociation free energy and acidity equilibrium constant values in water phase for—**A1** (AFB_1_O_7_-H^+^), **A2** (AFB_1_O_10_-H^+^), **A3** (AFB_1_O_13_O_12_-H^+^), **A4** (AFB_1_O_12_-H^+^), **A5** (AFB_1_O_14_-H^+^), **A6** (AFB_1_O_13_-H^+^), **A7** (AFB_1_O_12_-O_13_-H^+^).

AFB_1_H^+^ ⇆ AFB_1_ + H^+^				
Structure	ΔG Formation kcal/mol	ΔG Dissociation Reaction kcal/mol	Ka	pKa
**A1**	−694,497.342	−0.230	0.68	0.17
**A2**	−694,494.788	−2.324	50.53	−1.70
**A3**	−694,518.340	21.228	2.76 × 10^16^	15.56
**A4**	−694,515.571	18.459	2.95 × 10^14^	13.53
**A5**	−694,489.274	−7.838	556,713.56	−5.76
**A6**	−694,510.194	13.082	2.58 × 10^10^	9.59
**A7**	−694,513.410	16.298	1.13 × 10^12^	11.95

**Table 4 toxins-12-00283-t004:** Interaction energies between the protonated aflatoxin B_1_ (AFB_1_H^+^) molecule and CH_3_OH—A3-OHCH_3_ (AFB_1_-O_13_O_12_-H^+^∙∙∙OHCH_3_), A4-OHCH_3_ (AFB_1_O_12_H^+^∙∙∙OHCH_3_), A6-OHCH_3_ (AFB_1_O_13_H^+^∙∙∙OHCH_3_), and A7-OHCH_3_ (AFB_1_O_12_-O_13_-H^+^∙∙∙OHCH_3_).

Phase	Structure/Stability	A3-OHCH_3_	A4-OHCH_3_	A6-OHCH_3_	A7-OHCH_3_
Gas	Structure energy (kcal/mol)	−767,270.3	−767,265.1	−767,259.8	−767,269.4
	Interaction energy (kcal/mol)	−15.8	−10.7	−18.8	−19.9
Water	Structure energy (kcal/mol)	−767,310.6	−767,310.8	−767,305.8	−767,310.6
	Interaction energy (kcal/mol)	−7.7	−7.9	−12.0	−12.8

**Table 5 toxins-12-00283-t005:** Interaction energies between aflatoxin B_1_ (AFB_1_) protonated forms and protonated ammonium ion CH_3_NH_3_^+^. A3-^+^H_3_NCH_3_ (AFB_1_O_12_O_13_∙∙∙^+^H_3_NCH_3_), A4-^+^H_3_NCH_3_ (AFB_1_O_13_∙∙∙^+^H_3_NCH_3_), A6-^+^H_3_NCH_3_ (AFB_1_O_12_∙∙∙^+^H_3_NCH_3_) and A7-^+^H_3_NCH_3_ (AFB_1_O_13_O_12_∙∙∙^+^H_3_NCH_3_).

Phase	Structure/Stability	A3-^+^H_3_NCH_3_	A4-^+^H_3_NCH_3_	A6-^+^H_3_NCH_3_	A6-^+^H_3_NCH_3_
Gas	Structure energy (kcal/mol)	−754,981.1	−754,979.8	−754,976.5	−754,977.8
	Interaction energy (kcal/mol)	27.8	29.1	27.4	26.1
Water	Structure energy (kcal/mol)	−755,123.2	−755,120.4	−755,115.5	−755,118.8
	Interaction energy (kcal/mol)	0.3	−1.2	−5.3	-8.6

**Table 6 toxins-12-00283-t006:** Interaction energies between AFB_1_ (AFB_1_H^+^) and CH_3_COO^−^. A3-^−^OOCCH_3_ (AFB_1_O_13_O_12_-H^+^∙∙∙^−^OOCCH_3_), A4-^−^OOCCH_3_ (AFB_1_O_12_H^+^∙∙∙∙∙∙^−^OOCCH_3_), A6-^−^OOCCH_3_ (AFB_1_O_13_H^+^∙∙∙^−^OOCCH_3_) and A7-^−^OOCCH_3_ (AFB_1_O_12_O_13_-H^+^∙∙∙^−^OOCCH_3_).

Phase	Structure/Stability	A3-^−^OOCCH_3_	A4-^−^OOCCH_3_	A6-^−^OOCCH_3_	A7-^−^OOCCH_3_
Gas	Structure energy (kcal/mol)	−838,180.2	−838,181.1	−838,180.1	−838,177.7
	Interaction energy (kcal/mol)	−124.5	−125.3	−137.8	−126.9
Water	Structure energy (kcal/mol)	−838,197.7	−838,198.0	−838,197.2	−838,196.9
	Interaction energy (kcal/mol)	−31.7	−32.0	−40.2	−36.0

**Table 7 toxins-12-00283-t007:** Interaction energies between protonated aflatoxin B_1_ (AFB_1_) and O=C(CH_3_)_2_. A3-O=C(CH_3_)_2_ (AFB_1_O_13_O_12_-H^+^∙∙∙O=C(CH_3_)_2_), A4-O=C(CH_3_)_2_ AFB_1_O_12_-H^+^∙∙∙O=C(CH_3_)_2_), A6-O=C(CH_3_)_2_ (AFB_1_O_13_-H^+^∙∙∙O=C(CH_3_)_2_) and A7-O=C(CH_3_)_2_ AFB_1_O_12_O_13_-H^+^∙∙∙O=C(CH_3_)_2_).

Phase	Structure/Stability	A3-O=C(CH_3_)_2_	A4-O=C(CH_3_)_2_	A6-O=C(CH_3_)_2_	A7-O=C(CH_3_)_2_
**Gas**	Structure energy (kcal/mol)	−815,867.5	−815,867.3	−815,862.2	−815,865.0
	Interaction energy (kcal/mol)	−14.0	−18.9	−22.1	−16.6
**Water**	Structure energy (kcal/mol)	−815,908.0	−815,910.0	−815,905.2	−815,903.0
	Interaction energy (kcal/mol)	−5.2	−7.3	−11.4	−5.4

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
