# Peer review of "A Theoretical Study of the Adsorption Process of B-aflatoxins Using Pyracantha koidzumii (Hayata) Rehder Biomasses"

_toxins, 2020, doi:10.3390/toxins12050283_

Round 1
Reviewer 1 Report
Major comments: The connection between the material of biosorption (Pyracantha koidzumii, Hayata) and the prediction strategy used in this study is weak. Although the tested function groups were defined based on the chemical properties of Hayata, there could be many other phytochemical materials that are capable of interacting with AFB1 via hydroxyl group, amino group, and carboxyl group as well. If the current study was trying to unveil the possible mechanisms behind the biosorption specifically on Hayta, the conclusion is still vague and more validation from the wet lab will be needed.
Minor comments:
- Line 20, and 30: in vivo and in vitro, please be italics. Please check the whole article.
- Line 28: ‘Kurtzman et al.,’ should be deleted.
- Line 57, and 63: Pyracantha koidzumii need italics.
- Tables 1 and 5: Please explain the rationale why we need to analyze the energy values and interaction energies of the protonated structures of the AFB1 optimized molecule in the gas phase? Mycotoxins usually found in solid and liquid phases.
- Table 1 and 5: Please explain why the â–³E kcal/mol of A3 were 0 in water solution and gas phase?
- Line 326: 3.3 corrected to be 2.4.
- The section of the discussion is missing.
- Please add more detail about the materials and methods regarding the analysis in gas-phase, and water solution
Author Response
"Please see the attachment."

Reviewer 2 Report
Nice theoretical study with practical applications, I only found small mistakes
Line 57, 63 - all Latin names should be italic
Round 2
Reviewer 1 Report
The authors have answered questions and revised the manuscript properly. The present version is suitable for publication.